# Three-Dimensional Morphological Measurement Method for a Fruit Tree Canopy Based on Kinect Sensor Self-Calibration

**Haihui Yang** [1], **Xiaochan Wang** [1,2] **and Guoxiang Sun** [1,2,*]

1   College of Engineering, Nanjing Agricultural University, Nanjing 210031, China;
    yanghaihui131@163.com (H.Y.); wangxiaochan@njau.edu.cn (X.W.)
2   Jiangsu Province Engineering Lab for Modern Facility Agriculture Technology & Equipment,
    Nanjing 210031, China
*   Correspondence: sguoxiang@njau.edu.cn; Tel.: +86-25-5860-6585

**Abstract:** Perception of the fruit tree canopy is a vital technology for the intelligent control of a modern standardized orchard. Due to the complex three-dimensional (3D) structure of the fruit tree canopy, morphological parameters extracted from two-dimensional (2D) or single-perspective 3D images are not comprehensive enough. Three-dimensional information from different perspectives must be combined in order to perceive the canopy information efficiently and accurately in complex orchard field environment. The algorithms used for the registration and fusion of data from different perspectives and the subsequent extraction of fruit tree canopy related parameters are the keys to the problem. This study proposed a 3D morphological measurement method for a fruit tree canopy based on Kinect sensor self-calibration, including 3D point cloud generation, point cloud registration and canopy information extraction of apple tree canopy. Using 32 apple trees (Yanfu 3 variety) morphological parameters of the height (H), maximum canopy width (W) and canopy thickness (D) were calculated. The accuracy and applicability of this method for extraction of morphological parameters were statistically analyzed. The results showed that, on both sides of the fruit trees, the average relative error (ARE) values of the morphological parameters including the fruit tree height (H), maximum tree width (W) and canopy thickness (D) between the calculated values and measured values were 3.8%, 12.7% and 5.0%, respectively, under the V1 mode; the ARE values under the V2 mode were 3.3%, 9.5% and 4.9%, respectively; and the ARE values under the V1 and V2 merged mode were 2.5%, 3.6% and 3.2%, respectively. The measurement accuracy of the tree width (W) under the double visual angle mode had a significant advantage over that under the single visual angle mode. The 3D point cloud reconstruction method based on Kinect self-calibration proposed in this study has high precision and stable performance, and the auxiliary calibration objects are readily portable and easy to install. It can be applied to different experimental scenes to extract 3D information of fruit tree canopies and has important implications to achieve the intelligent control of standardized orchards.

**Keywords:** three-dimensional reconstruction; Kinect; three-dimensional point cloud; fruit tree canopy

## 1. Introduction

The canopy is the first part of fruit trees to contact with light and the outside environment; it is also the main place to carry out respiration and photosynthesis. Studies of information perception show that the canopy determines the growth of the fruit tree, and then affects the yield and economic benefits [1,2]. In order to make comprehensive research of canopies, we can reconstruct three-dimensional (3D) objects of the real world digitally in a computer, and generate virtual 3D models [3,4]. To promote the automation and information construction of the orchard, 3D reconstruction of plants can be applied to



the fruit tree canopy. Canopy morphology and structure are visualized on computer, and relevant parameter information such as canopy height and volume are extracted by the developed object-based image analysis (OBIA) algorithm. These data can provide theoretical basis and technical support for the intelligent control, monitoring, and management of orchards [5,6].

In the field of 3D reconstruction for fruit trees, Sansh and Rosll (2013) used a two-dimensional (2D) terrestrial laser radar scanner (2DTLS) to establish a 3D dynamic measurement system [7,8]. In their work, a light detection and ranging (LIDAR) system was used to scan trees on both sides, and the images from the two angles were merged to generate a 3D point cloud from which the fruit tree canopy information was extracted. Nguyen et al. (2016) used an RGB-D(red, green, blue and depth) camera to design image reconstruction that includes color information and 3D shape information [9], and based on the apple colors and shapes, the fruits on the canopy were accurately detected and located; this algorithm is suitable for robot picking and can estimate the output yield of an orchard. Narváez (2016) used a drone to capture large-area orchards' thermal infrared images, which were merged with laser LIDAR data for 3D reconstruction [10], providing large-area 3D visual images in the Global Navigation Satellite System (GNSS) environment for multiple orchards. Torres-Sánchez (2018) proposed an OBIA algorithm for measuring the 3D geometric features of apricot trees, in which a low-cost RGB camera carried in a drone was used to generate a measured point cloud followed by technical analysis of the image [11]. In China, Yan (2000) and Hu (2014) used a multiple-image modeling technique for achieving reconstruction and used a 90°-difference image sequence to interactively adjust the branch position information for obtaining a 3D model of fruit trees [12]. Sun et al. (2012) used FastSCAN based on laser scanning to obtain the point cloud data of plant leaves and obtained 3D models of various plant leaves by triangular meshing after preprocessing [13]. Sun et al. (2019) used an autonomous Kinect v2 sensor position calibration to reconstruct 3D point clouds for high-throughput plant phenotyping analysis and to extract the morphological parameters of plants [14]. Chen et al. (2016) used a Microsoft Kinect sensor to capture 3D point cloud data. A viewpoint feature histogram (VFH) descriptor for the 3D point cloud data then encodes the geometry and viewpoint, so an object can be simultaneously recognized and registered in a stable pose, and the information is stored in a database [15].

In the above reconstruction methods, one generated by an unmanned aerial vehicle (UAV, drone, Trimble UX5, Trimble Navigation, Sunnyvale, CA, USA) equipped with cameras can perceive a large area of multiple fruit trees [16], but the detection of the fruit tree details is not accurate enough.

In the current research, 2D laser radar and a monocular camera cannot provide comprehensive information for the perception, the accuracy of using drones and laser radars cannot guide the practical production, and the 3D point cloud established by RGB-D mainly focuses on the local branches under a single visual angle, rendering the extracted information not comprehensive enough.

Moreover, due to the complex environment of the actual orchard, there are many factors affecting the measurement, and the information using a single visual angle cannot meet the required accuracy of the information extraction; information merging under multiple visual angles is thus needed. Therefore, the self-calibration of sensor poses under multiple visual angles is the key issue to solve in canopy 3D reconstruction for fruit trees [17,18].

In order to solve the fruit tree 3D reconstruction problem, this study proposed a fast 3D reconstruction method based on RGB-D image self-calibration under the multiple-visual-angle mode. A Kinect sensor was used for the self-calibration of poses and to quickly match RGB-D at multiple visual angles for achieving 3D point cloud reconstruction. In this study, using apple trees (Yanfu 3 variety) as the measurement object, manual measurements were compared with the morphological parameters extracted from the 3D reconstruction calculation, and the relative error and correlation were analyzed to provide a theoretical basis and support for fruit tree canopy 3D morphological information extraction and measurement.

## 2. Materials and Methods

### 2.1. Structure and Principle of the Measurement System

The measurement system of this method was composed mainly of a Kinect sensor, a graphic workstation, a tripod, red, yellow and blue calibration balls (10 cm in diameter), and a 220 V mobile power supply. The Kinect sensor (Kinect-v2, Microsoft, Redmond, WA, USA), used the version of Kinect for Windows 2.0, consisting of a color camera and a depth sensor. The RGB and depth images had resolutions of 1920 × 1080 px and 640 × 480 px, respectively. The frame rate was 30 fps, and the measurement distance range of the target objects in this study was 170–320 cm. The tripod was used in combination with the Kinect sensor for adjusting and fixing the angles and distance measurement during the fruit tree measurements. The RYB calibration balls (one of each) were placed around the target object for the self-calibration of the poses. The graphics workstation had an Intel core i5-2520M processor, Windows 10 64-bit operating system, 4GB ECC RAM, and an NVIDIA GeForce GT 635M graphics card. The software environment was a mixed programming environment (Visual Studio 2015 and MATLAB 2016a), and the software developed for the fruit tree canopy measurement was based on Kinect2 SDK and C++ wrapper functions for Microsoft Kinect 2. Figure 1 shows a schematic illustration of the measurement system structure. Specifically, Figure 1a shows a color image of the fruit trees captured by Kinect and Figure 1b shows the image after depth and color matching for the fruit trees.

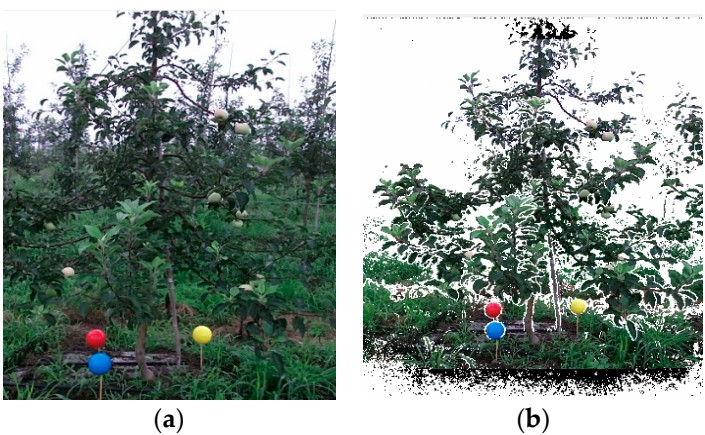

(**a**)　　　　　　　　　　　(**b**)

**Figure 1.** Schematic illustration of the measurement system structure: (**a**) color image captured; (**b**) image after depth and color matching.

The measurement procedure of the fruit tree canopy 3D morphological measurement system based on Kinect sensor self-calibration is shown in Figure 2. The first step was to initialize the measurement parameters. An appropriate camera shooting position was selected considering the actual scene and terrain conditions. The key factors included the measurement angle, measurement distance and camera height (the measurement angle choice captured the whole fruit tree within the camera's field of view, the measurement distance was 150–350 cm and the camera was placed at half the height of a given fruit tree). The RYB balls were placed around the trunk and it was verified that the three balls were not obscured at any angle when the fruit tree was shot by the cameras. The second step was to acquire fruit tree RGB-D images under multiple visual angles. According to the requirements and the actual conditions, the required number of visual angles was determined. A nonfixed angle was adopted in the visual angle acquisition (as long as it was verified that the visual angle acquisition can meet the panoramic recovery requirements) and RGB-D images were then acquired by turns under each visual angle. In this paper, we adopted two single visual angle measurement modes (V1 and V2) and a double visual angle measurement mode (with V1 and V2 merged). The third step was to generate the 3D point cloud for a single visual angle. Using the Kinect camera's internal parameters,

the single-visual-angle RGB-D images were converted into a 3D point cloud image. The fruit tree bounding box segmentation and denoising were performed for the 3D point cloud. The fourth step was to reconstruct the fruit tree 3D point cloud. The point cloud coordinates of the three balls under a single visual angle were identified. According to the spherical coordinates of these calibration balls at different angles, the corresponding spatial transition matrix was obtained. The matrix was applied to the fruit tree point cloud under multiple visual angles to achieve the unification of the coordinate system, thereby completing the coarse matching of the fruit tree 3D point cloud. Finally, the iterative closest point (ICP) algorithm was used for self-registration to obtain the complete model of the fruit tree 3D point cloud.

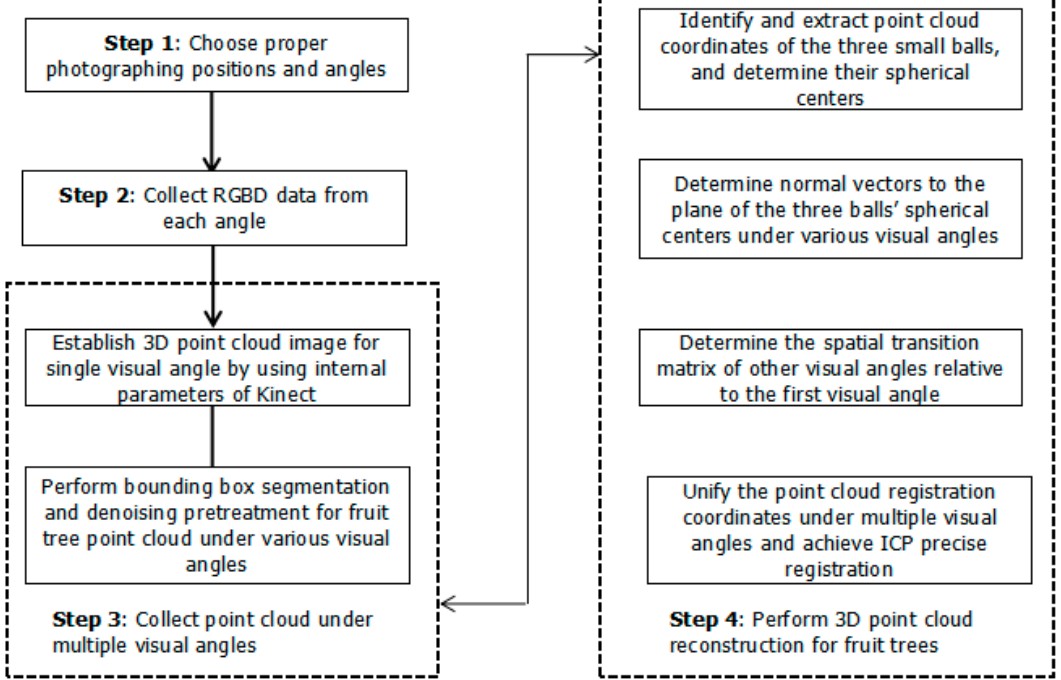

**Figure 2.** Diagram of three-dimensional (3D) reconstruction methodology.

*2.2. Procedure of 3D Reconstruction*

The first step is to collect the RGB-D image for each visual angle. Considering the fruit tree planting distance in the actual orchard, two visual angles were chosen for the reconstruction. According to the Kinect sensor's internal parameters, the 3D point cloud image of a single visual angle was converted. The bounding box was used to remove the interfering pixels of the surrounding environment, and outlier denoising was used to cancel out the unrelated noise, thereby generating a preprocessed single-visual-angle 3D point cloud image. Figure 3 illustrates the 3D reconstruction of two visual angles of the fruit tree; point cloud coordinates are set as PointCloud1 and PointCloud2.

According to the bounding box of each ball and the threshold of color, the point cloud coordinates of the RYB balls were segmented and identified. Figure 4 shows the specific identification results, the numbers of red ball points were 127 and 161, yellow were 143 and 132, and blue were 193 and 82. The numbers of points can thus satisfy the subsequent processing requirements. The spherical coordinates R1 $(x, y, z)$, Y1 $(x, y, z)$, B1 $(x, y, z)$, R2 $(x, y, z)$, Y2 $(x, y, z)$, and B2 $(x, y, z)$ for the red, yellow, and blue balls under each visual angle were calculated. Next, according to the obtained coordinate values for R1, Y1, B1, and R2, Y2, B2, the formula M $(x, y, z)$ = mean (R, Y, B) was used to calculate the centers of gravity M1 $(x, y, z)$ and M2 $(x, y, z)$ of the triangle determined by the three ball centers in each angle. According to the three points R1, Y1, and B1, plane s1 was determined, and according to R2, Y2, and B2, plane s2 was determined. Thus, point M1 was determined on plane s1, and point M2 was determined on plane s2. Taking points M1 and M2 as the respective vertices for plane s1 and

plane s2, their corresponding normal vectors p1 (*a*, *b*, *c*) and p2 (*a*, *b*, *c*) were calculated. Moving points M1 and M2 to the origin O (0, 0, 0) in the coordinate system and translating the vectors p1 and p2 along with M1 and M2 at the same time, the apexes of p1 and p2 would both be at O (0, 0, 0) after the translation; these were then normalized and denoted as p3 and p4. According to the point cloud translation equation PointCloud' = PointCloud − M, the PointCloud1 and PointCloud2 point cloud coordinates were translated to obtain PointCloud3 and PointCloud4, respectively. Figure 5 shows the cloud images PointCloud3 and PointCloud4, with point M translated to the origin O (0, 0, 0) for each visual angle point and the normal vectors p3 and p4 that were normal to the plane of the ball centers of the three colored balls.

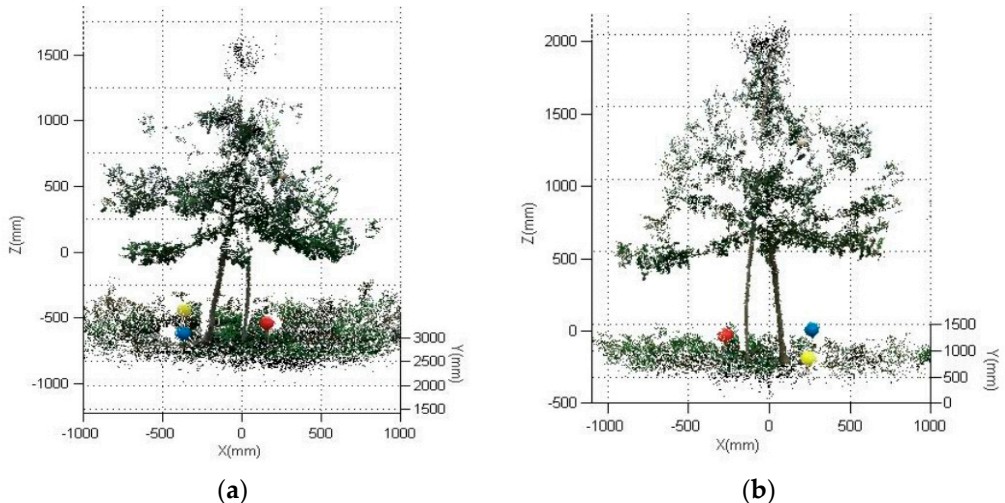

(**a**)                    (**b**)

**Figure 3.** Point cloud image for a fruit tree under different visual angles: (**a**) point cloud coordinate: PointCloud1; (**b**) point cloud coordinate: PointCloud2.

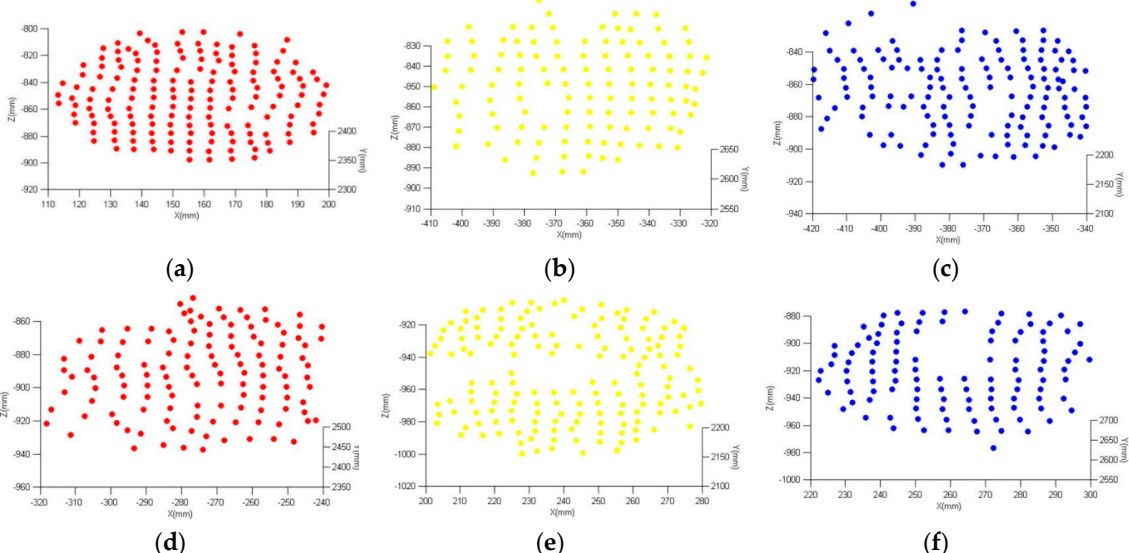

(**a**)                (**b**)                (**c**)

(**d**)                (**e**)                (**f**)

**Figure 4.** Identification diagram with calibration balls: (**a**) red ball point cloud from the first visual angle; (**b**) yellow ball point cloud from the first visual angle; (**c**) blue ball point cloud from the first visual angle; (**d**) red ball point cloud from the second visual angle; (**e**) yellow ball point cloud from the second visual angle; (**f**) blue ball point cloud from the second visual angle.

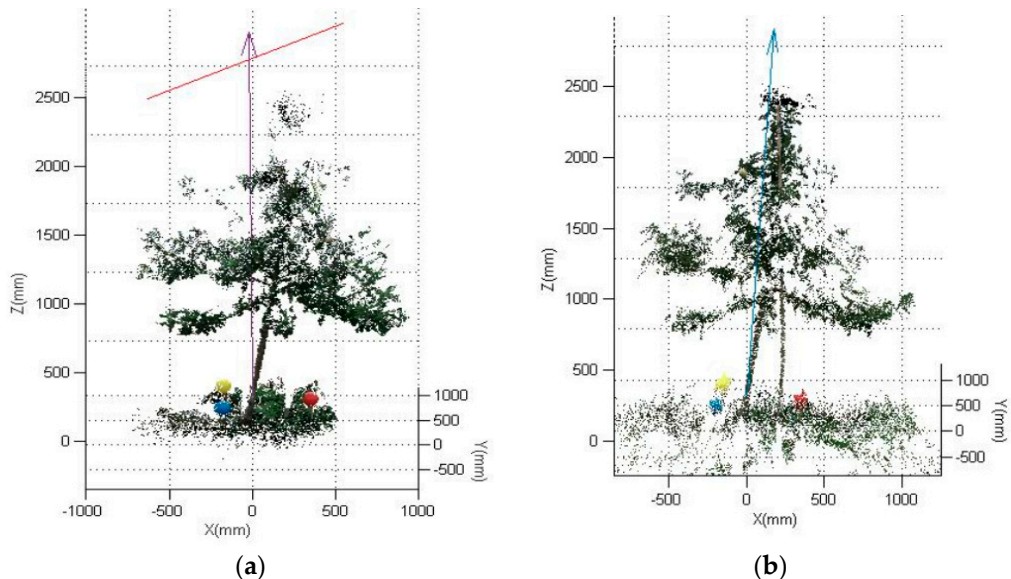

**Figure 5.** Schematic diagram of normal vectors p3 and p4 to plane of calibration ball spherical centers: (**a**) schematic diagram of normal vectors p3; (**b**) schematic diagram of normal vectors p4.

Rotating the normal vectors p3 and p4 to the Z axis (0, 0, 1) of the Kinect coordinate system and determining the spatial transformation rotation matrices of p3 and p4, Rx1, Ry1 and Rx2, and Ry2, respectively, were calculated. The calculation process was as below: first, the vectors p3 and p4 were rotated α degrees around the X axis to the XOY plane, with the corresponding rotation matrix being $R_x$ (α) (as shown in Equation (1)), and then they were rotated β degrees around the Y axis to the Z axis, with the corresponding rotation matrix being $R_y$ (β) (as shown in Equation (2)), i.e., z (0, 0, 1) = pi × $R_x$ (α)× $R_y$ (β). According to the obtained $R_x$ (α) and $R_y$ (β), the coordinates $Y_1'$ and $Y_2'$ after the translation operation with M for points Y1 and Y2 were connected with the origin O. Likewise, for vectors $OY_1'$ and $OY_2'$, the rotation matrix operations were performed for $R_x$ (α) and $R_y$ (β), and two vectors NOY1 and NOY2 were obtained (with the O as the vertex and with the vectors $OY_1''$ and $OY_2''$ as the directions). Finding the size of angle γ between the two vectors NOY1 and NOY2 at this point and substituting γ into Equation (3), the rotation matrix $R_z$ (γ) around the Z axis can be calculated. According to Equation (4), the point clouds upon the spatial transformation for the other visual angles relative to the first visual angle can be obtained.

The point cloud of the first visual angle, along with the point clouds of the other visual angles that were subjected to the above spatial transformation, was displayed in the same point cloud coordinate system, and the 3D point cloud image after coarse registration was obtained (Figure 6).

$$R_x(\alpha) = \begin{bmatrix} 1 & 0 & 0 \\ 0 & cos\alpha & -sin\alpha \\ 0 & sin\alpha & cos\alpha \end{bmatrix} \tag{1}$$

$$R_y(\beta) = \begin{bmatrix} cos\beta & sin\beta & 0 \\ -sin\beta & cos\beta & 0 \\ 0 & 0 & 1 \end{bmatrix} \tag{2}$$

$$R_z(\gamma) = \begin{bmatrix} cos\gamma & 0 & sin\gamma \\ 0 & 1 & 0 \\ -sin\gamma & 0 & cos\gamma \end{bmatrix} \tag{3}$$

$$PointCloud'_m(x, y, z) = PointCloud_m(x, y, z) \times R_x(\alpha) \times R_y(\beta) \times R_z(\gamma) \tag{4}$$

where: *PointCloud$_m$* is point cloud of the view m; (*x, y, z*) is point cloud coordinates; *PointCloud'$_m$* is point cloud scattered after transformation of the second view space; $R_x$ (*α*) is rotation matrix of the normal vector to XOY plane, $R_y$ (*β*) is rotation matrix of the normal vector to Z axis; and $R_z$ (*γ*) is the rotation matrix around Z axis for angle *γ* between the two vectors NOY1 and NOY2.

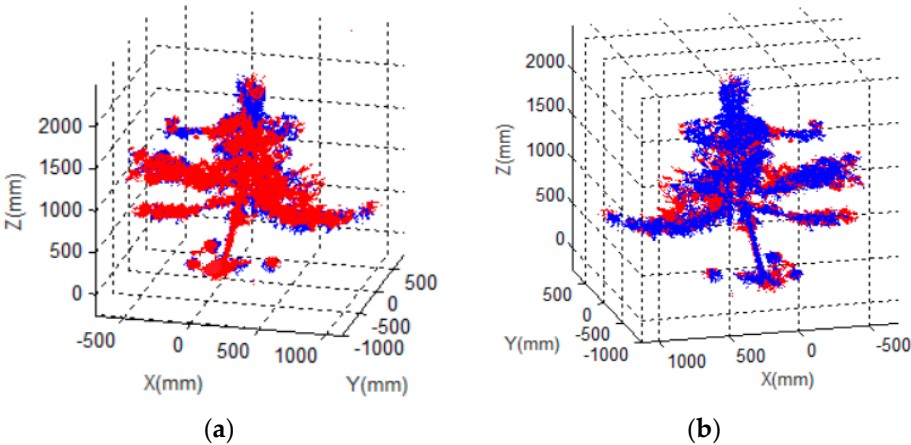

(**a**)　　　　　　　　　　　　　　　　　　　　(**b**)

**Figure 6.** 3D point cloud image after coarse registration: (**a**) point cloud with double-visual-angle coarse registration (visual angle 1); (**b**) point cloud with double-visual-angle coarse registration (visual angle 2).

The point cloud image after the coarse registration already had the basic parameters and shape of a fruit tree. However, due to the inherent error in the Kinect acquisition and the calculation error in the calibration process, the coarse registration cannot guarantee accurate information on the point cloud details, and further high-precision fine-tuning registration is needed. According to the ICP algorithm, an iterative registration algorithm proposed by Besl and McKay (1992) [19], free surfaces and curves can be registered, as well as point clouds. Moreover, the ICP algorithm does not need the accurate information of the initial calibration reference point or the calibration point; it uses iteration to find the nearest point for all the points in the desired registration point cloud, and then continues to calculate the spatial transition matrix by using these known points as standard points until the preset number of times is reached or the set convergence threshold condition is met. Figure 7 is a model of the fruit tree 3D point cloud after precise registration via ICP, containing accurate 3D information (HWD) about the fruit tree. At this point, the fruit tree 3D reconstruction is completed.

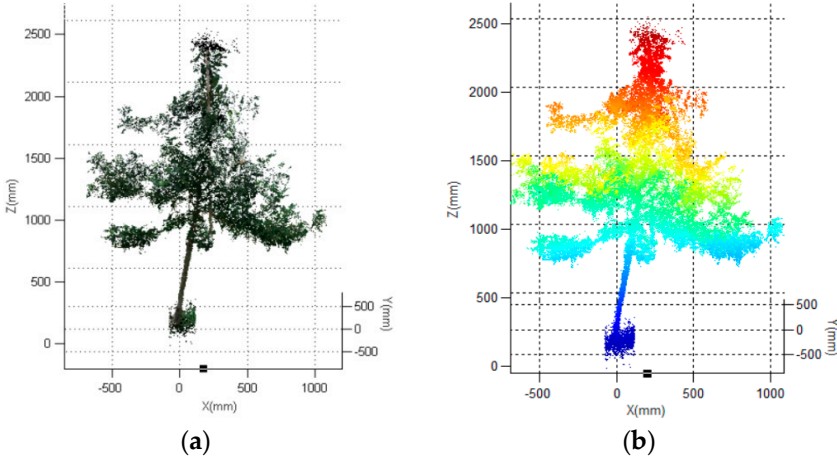

(**a**)　　　　　　　　　　　　　　　　　　　　(**b**)

**Figure 7.** Model of the fruit tree 3D point cloud after precise registration via iterative closest point (ICP): (**a**) 3D point cloud of fruit tree after ICP (image with original color); (**b**) 3D point cloud of fruit tree after ICP (image with coloration).

## 2.3. Calculation Method for the 3D Morphology of the Fruit Tree Canopy

According to the 3D point cloud model of the fruit tree, the fruit tree canopy height (H), the canopy maximum tree width (W) and the canopy thickness (D) were calculated, where D is the length of canopy in the vertical direction. The actual data obtained by measured measurements were compared with the calculated values and an error analysis was conducted. Since the rotation matrix in the 3D reconstruction was processed by normalization, the distance values in the reconstructed point cloud graph coordinate system would correspond to the actual value. The Z-axis coordinate of the highest point of the canopy was extracted and was compared with the Z-axis coordinate of the ground to obtain the difference, and the fruit tree height H can thereby be obtained. The specific calculation formula is shown in Equation (5), and the schematic diagram is shown in Figure 8a.

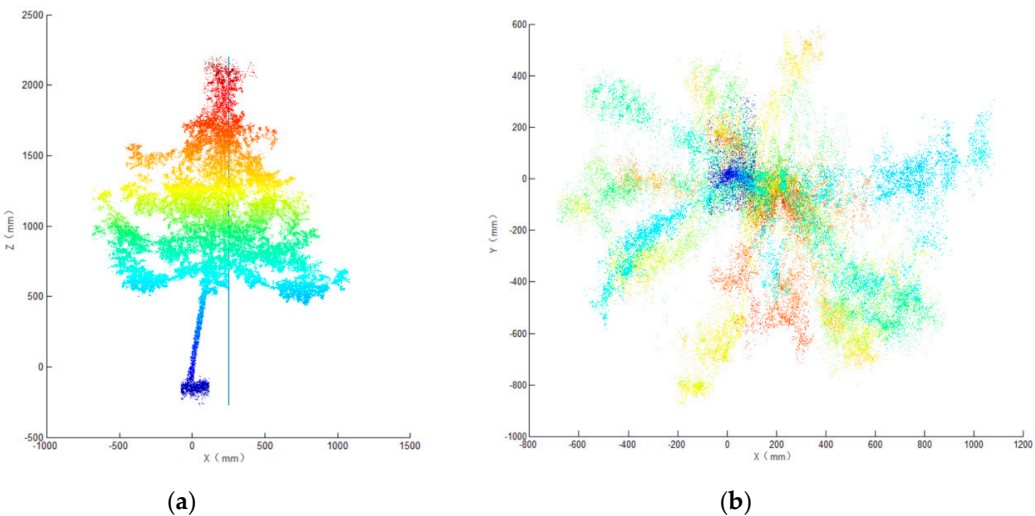

|   (a)   |   (b)   |

**Figure 8.** Schematic diagram of height and maximum width of fruit tree: (**a**) schematic diagram for the height (H); (**b**) schematic diagram for the maximum tree width (W).

While the coverage volume of the fruit tree canopy is large, branches overlapped and both located in the main part of the point cloud, W would be different under different visual angles. The study chose the point cloud to make a projection under the XOY visual angle, and by traversing the maximum distance between the two points under the projection, the maximum value would be the maximum W of the fruit tree canopy. The specific calculation formula is shown in Equation (6), and the schematic diagram is shown in Figure 8b.

To identify the fruit tree canopy, in this study, kernel smoothing was performed for the Z-axis point cloud quantity distribution of the 3D point cloud after the fruit tree reconstruction, with the calculation formula shown in Equation (7). The distribution chart of the probability density $f$ is shown in Figure 9. According to the statistics, setting the threshold as $3 \times 10^{-4}$ and setting the probability density greater than the threshold for the first time as the minimum value of the canopy height, the difference between the $Z_i$ coordinate corresponding to the threshold and the fruit tree top $Z_{max}$ was calculated. The thickness of the fruit tree canopy can be obtained, with the calculation formula shown as Equation (8).

$$H = Z_{max} - Z_{ground} \tag{5}$$

$$W = \max\left(\sum_{i=1}^{num}\sum_{j=1}^{num}\sqrt{\left(\left(X_i - X_j\right)^2 + \left(Y_i - Y_j\right)^2\right)}\right) \tag{6}$$

$$f(z) = \frac{1}{nh}\sum_{i=1}^{n} K\left(\frac{Z - Zi}{h}\right) \tag{7}$$

$$D = Z_{max} - Z_{f=0.0003} \tag{8}$$

where: $Z$ is the vertical axis value of the point cloud of a fruit tree; $x$ and $y$ are the corresponding coordinates of the point cloud under the XOY projection; $n$ is the total number of samples; $h$ is the band width; and $K$ is the kernel smoothing function.

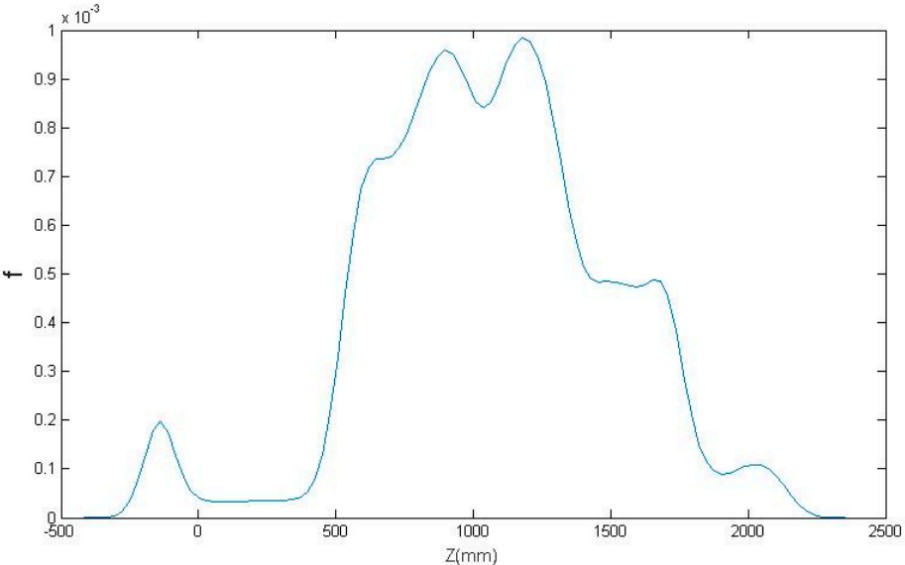

**Figure 9.** Probability density distribution on the Z-axis.

## 3. Results

### 3.1. Data Analysis of 3D Morphological Measurements for Fruit Tree Canopy

Due to the photographing distance limits in the actual experimental scene, information of two visual angles was selected for the reconstruction. Figure 10 shows part of the reconstruction for the fruit tree point cloud. According to the above algorithm, relevant information was extracted and was compared with the manual measurement values to assess the errors in the application in the orchard base. For the 3D point cloud after reconstruction, the coefficient of determination ($R^2$), root mean square error (RMSE), and average relative error (ARE) values between the extracted information and the manual measurements for H, W, and D under three visual angles (V1, V2, and V1 and V2 merged) were statistically analyzed.

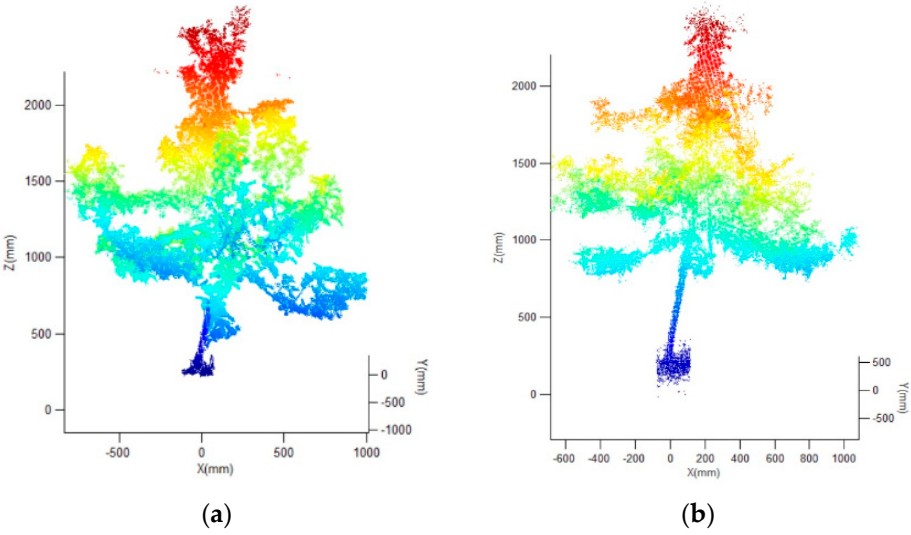

(a)　　　　　　　　　　　　　　　　　　　　　(b)

**Figure 10.** Part of 3D point cloud reconstruction for fruit trees: (**a**) 3D point cloud reconstruction for fruit tree a; (**b**) 3D point cloud reconstruction for fruit tree b.

Figure 11 shows the data comparison between the manually measured values and the calculated values for the fruit tree canopy W, H and D for the samples. In the calculation of the height H and the canopy thickness D, errors occurred in the calculation process primarily due to the uneven ground surface in the actual scene, and for the lack of data at the highest point during the photographing for some samples. When calculating the tree width W, the calculated values were generally large, mainly due to the complexity of the actual scene of the orchard (light, flying insects, dust, etc.) that resulted in some noise.

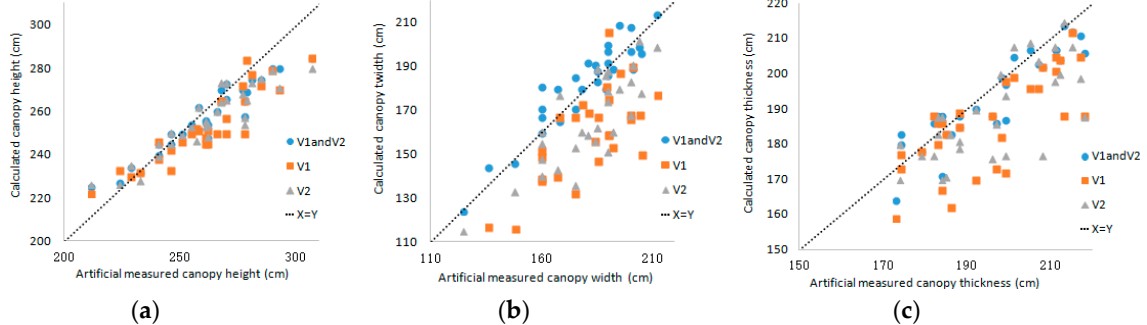

| (a) | (b) | (c) |

**Figure 11.** Measured and calculated values for H, W and D: (**a**) data comparison between the manually measured values and the calculated values for the fruit tree canopy H; (**b**) data comparison between the manually measured values and the calculated values for the fruit tree canopy W; (**c**) data comparison between the manually measured values and the calculated values for the fruit tree canopy D.

### 3.2. Error Analysis of 3D Morphological Measurements for Fruit Tree Canopy

Since 32 fruit tree samples were reconstructed under three modes (single visual angle V1, single visual angle V2, and double visual angles V1 and V2 merged). The mean values of the coefficient of determination ($R^2$), RMSE (root mean square error) values and ARE (average relative error) values for H, W, and D between calculation and the manual measurement were shown in Table 1.

**Table 1.** Relationship between calculated and measured values of morphology for sample fruit trees. RMSE, root mean square error; ARE, average relative error; $R^2$, coefficient of determination.

| Parameter | Visual Angle | $R^2$ | RMSE | ARE |
|---|---|---|---|---|
| | V1 | 0.91 | 12.28 | 3.8% |
| Height (H) | V2 | 0.84 | 13.79 | 3.3% |
| | V1 and V2 | 0.96 | 8.72 | 2.5% |
| | V1 | 0.86 | 26.09 | 12.7% |
| Width (W) | V2 | 0.91 | 20.27 | 9.5% |
| | V1 and V2 | 0.97 | 9.71 | 3.6% |
| | V1 | 0.73 | 13.38 | 5.0% |
| Thickness (D) | V2 | 0.67 | 14.34 | 4.9% |
| | V1 and V2 | 0.82 | 9.12 | 3.2% |

The analysis shows that the calculated values of the fruit tree height, the maximum tree width, and the canopy thickness were significantly correlated with their corresponding measured values. The calculation accuracy was improved under the double visual angle mode compared to the single visual angle mode, and the measurement superiority for the maximum tree width was particularly significant. This was mainly because the single visual angle mode could only provide information about a single visual angle of the camera, but was not able to provide comprehensive 3D information, making its tree width measurement error relatively large. In contrast, the 3D information under which the double visual angles were merged was relatively comprehensive, and thus its calculation error for the tree width was relatively small.

## 4. Discussion

This study solved the fast registration problem for a multiple-visual-angle RGB-D point cloud, the registration method is very simple to install and disassemble, it can also be used in complex orchards to calculate the relevant parameters and extract more specific information.

The 3D point cloud reconstruction method based on Kinect self-calibration proposed in this study has a comprehensive display of fruit tree morphological information and high precision of parameter extraction, and the auxiliary calibration objects are readily portable and easily installed. The method can be applied to different experimental scenes to extract 3D information of a fruit tree canopy and has important implications to achieve the intelligent control of a standardized orchard.

While this method could reconstruct the fruit tree canopy shape and extract the related parameters, there are still insufficiencies that deserve improvement. The acquisition conditions of the infrared camera are harsh in the outdoor environment, so accurate information could only be collected in the morning or evening, and the Kinect can be further explored so that it can be used more widely. This method can also be used in the automatic picking robot when it can accurately identify the position and number of fruit. In the future, more calibration balls or more accurate calibration objects could be used to improve the experiment, and the space transformation matrix could be more accurate, so as to improve the reconstruction effect and extract parameters more accurately.

## 5. Conclusions

This study proposed a 3D morphological measurement method for a fruit tree canopy based on Kinect sensor self-calibration, including 3D point cloud generation, point cloud registration and canopy information extraction of apple tree canopy. The results showed that on both sides of the fruit trees, the average relative error (ARE) values of the morphological parameters including the fruit tree height (H), maximum tree width (W) and canopy thickness (D) between the calculated values and measured values were 3.8%, 12.7% and 5.0%, respectively; under the V1 mode, the ARE values under the V2 mode were 3.3%, 9.5% and 4.9%, respectively; and the ARE values under the V1 and V2 merged mode were 2.5%, 3.6% and 3.2%, respectively. The measurement accuracy of the tree width (W) under the double visual angle mode had a significant advantage over that under the single visual angle mode.

The 3D point cloud reconstruction method based on Kinect self-calibration proposed in this study has high precision and stable performance, and the auxiliary calibration objects are readily portable and easy to install. It can be applied to different experimental scenes to extract 3D information of fruit tree canopies and has important implications to achieve the intelligent control of standardized orchards.

**Author Contributions:** Conceptualization, G.S. and X.W.; methodology, G.S. and X.W.; software, G.S. and H.Y.; validation, G.S., and H.Y.; formal analysis, G.S. and H.Y.; investigation, G.S., X.W., and H.Y.; writing—original draft preparation, G.S. and H.Y.; writing—review and editing, G.S., X.W., and H.Y.; project administration, X.W. and H.Y.; funding acquisition, G.S., X.W.

**Funding:** This research was funded by the National Key R&D Program of China (Grants No. 2017YFD0701400).

**Acknowledgments:** The authors would like to thank Liqun Wei and Fengjie Wang for their technical and research support.

**Conflicts of Interest:** The authors declare no conflict of interest.

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
