# Peer review of "Three-Dimensional Morphological Measurement Method for a Fruit Tree Canopy Based on Kinect Sensor Self-Calibration"

_agronomy, doi:10.3390/agronomy9110741_

Round 1

Reviewer 1 Report

I wish you well in your future research and will be looking for further improvements to the methodology. I also found the Kinect difficult to use in daylight.

I hope that the software used will become available at sometime.

There were just a couple of minor text changes that I found.

Reviewer 2 Report

the reviewed paper significantly improved with respect to the original one.

still some clarification are needed.

row 31 please use brackets for H, W and D

rows 33-35 please use the same number of decimal (also for table 1)

row 91, avoid repeating "reconstruction" in the same sentence

row 92 please remove "in combination"

row 126 as in the previous review, we ashed to better define Kinet sensor

row 56, 123 please the first time use acronyms you must indicated what they refer to

please be consistent all over the text for "observed" and "simulated". Sometimes we found "measured" and ""calculated"

Reviewer 3 Report

Please see comments on the printed, marked, scanned, and uploaded manuscript.

Author Response

This manuscript is a resubmission of an earlier submission. The following is a list of the peer review reports and author responses from that submission.

Round 1

Reviewer 1 Report

Overall the research was well thought out and conducted. The introduction, M & M and Results are thorough and ignoring English style and grammar for a moment, make sense to the reader.

The discussion was not well written and missed many important aspects as it only restated some of the results and and M & M.

Given the highly technical nature of the work involved it was disappointing to see that it had only been used to measure height, depth and width of the canopy- are these really the only metrics that can be derived from the point cloud?

Many other aspects of the work need to be discussed, e.g.:

relative cost between this sensor and others used Limitations to the approach, particularly in relation to automation. The need for the three coloured ball seems to me to be a significant drawback Other measures such as surface area and volume of the canopy could be examined with this data set

The grammar and style need to worked on as in many places it is difficult to read. I commend the authors for undertaking this in their non-native language, and would encourage them to take some extra time to have it reviewed more thoroughly before submission.

Examples of suggested edits
Page 1.

Line 10
Fruit tree canopy information perception is a vital technology for the intelligent control of a modern standardized orchard.

This might be written as:

Perception of the fruit tree canopy is a vital technology for the intelligent control of a modern standardized orchard.

Line 13: try to avoid putting the cart before the horse and avoid long sentences viz.

To perceive fruit tree canopy information efficiently and accurately in
complex orchard field environment, 3D fruit tree canopy information from different perspectives should be extracted and fused, the registration and fusion algorithm of canopy information from different perspectives and the extraction method of fruit tree canopy related parameters are the keys of the problem.

An alternative:
3D information from different perspectives must be combined in order to perceive the canopy information efficiently and accurately in complex orchard field environment. The algorithms used for registration and fusion of data from different perspectives and the subsequent extraction of fruit tree canopy related parameters are the keys to the problem.

Line 19 - split long sentences and remove unnecessary words
Using 32 apple trees (Yanfu 3 variety) as the measurement object, the morphological parameters including the height (H), maximum tree width (W) and canopy thickness (D) of the 32 fruit tree canopies were calculated, the accuracy and applicability of this method in morphological parameter extraction were statistically analyzed.

Using 32 apple trees (Yanfu 3 variety) morphological parameters the height
(H), maximum canopy width (W) and canopy thickness (D) were calculated. The accuracy and applicability of this method for extraction of morphological parameters were statistically analyzed.

Lines 26 and 27. V1 and V2 are introduced here and need to be explained or just refer to two modes and expand on this in M & maximum

First paragraph of the introduction needs recasting with the previous examples in mind as it is difficult to read.

These examples can be used throughout the paper as the English needs to be improved throughout.

Other comments
The introduction gets a bit repetitive (only in style, not content), but other than that it contains a lot of useful information and covers the literature.

Page 3:
Need more information on the Kinect Sensor - manufacturer, version. I don't think etc is good enough in line 101. What do I need to buy to do this myself?

Figure 1. The profile of the laptop and power supply can be dropped.

Line 129-130 I don't understand what is meant by the part in ().

Page 4.
Line 145 - by shown below do you mean Figure 3? Remember the journal will place figures where they see fit and it might not be below.

Page 5.
Line 159 - the number of red ball point clouds were 127... I think you are referring to the number of points in each cloud from the two sensor positions.

Page 7.
Line 219 and elsewhere PointCloud misspelt.

Page 8
Why is important to find the widest point rather than just the minimum and maximum y values?
Figure 8 (b) labels need to be larger, like 8(a). This needs to be checked on all figures

Page 11
Table 1. I think using r rather than R^2 is more appropriate when the analysis is correlation, besides you have two additional error measures.

Line 329 - replace "The statistical data..." with "The analysis..."

Line 331 The calculation accuracy was significantly improved... "significantly" would implies that you have tested this and I don't see any evidence of that.

Discussion
This lets the paper down as there are many aspects that not discussed and results are dropped in. You need to rethink this section from the start.

Reviewer 2 Report

This paper aim at presenting a method for measuring a tree canopy using a Kinect-sensor. Actually the issue is very interesting and the methodology promising, but I have to say that the poor English style, the confusing explanations and the discussion section that is more or less a summary of the paper, prevents the prompt publication of this paper.

Please, consider that the audience of Agronomy may be not ready for technicalities of 3D technologies and you must provide an efficient way to present your aim, your methods and your results.

Please, consider that Results and discussion section are poorly presented, with discussions that would benefit by introducing the performances/advantages of the proposed methodology with respect to other approaches. 

Please find in attach a commented PDF
